# Smart Radiotherapy Biomaterials for Image-Guided In Situ Cancer Vaccination

**DOI:** 10.3390/nano13121844

**Published:** 2023-06-12

**Authors:** Victoria Ainsworth, Michele Moreau, Romy Guthier, Ysaac Zegeye, David Kozono, William Swanson, Marian Jandel, Philmo Oh, Harry Quon, Robert F. Hobbs, Sayeda Yasmin-Karim, Erno Sajo, Wilfred Ngwa

**Affiliations:** 1Department of Radiation Oncology and Molecular Radiation Sciences, Johns Hopkins University, Baltimore, MD 21201, USA; mmoreau1@jhmi.edu (M.M.); hquon2@jhmi.edu (H.Q.); rhobbs3@jhmi.edu (R.F.H.); 2Department of Physics, Medical Physics, University of Massachusetts Lowell, Lowell, MA 01854, USAmarian_jandel@uml.edu (M.J.); erno_sajo@uml.edu (E.S.); 3Department of Radiation Oncology, Dana-Farber Cancer Institute, Harvard Medical School, Boston, MA 02115, USA; zegeye.y@northeastern.edu (Y.Z.); dkozono@bwh.harvard.edu (D.K.); syasmin-karim@bwh.harvard.edu (S.Y.-K.); 4Department of Cell and Molecular Biology, Northeastern University, Boston, MA 02115, USA; 5Department of Radiation Oncology, Weill Cornell Medicine, New York, NY 10065, USA; wis4004@med.cornell.edu; 6NanoCan Therapeutics Corporation, Princeton, NJ 08540, USA; poh@nanocan.life; 7Department of Radiation Oncology, Brigham and Women’s Hospital, Harvard Medical School, Boston, MA 02115, USA

**Keywords:** smart radiotherapy biomaterials, cancer, in situ vaccination, abscopal effect, image-guided radiotherapy, radio-immunotherapy

## Abstract

Recent studies have highlighted the potential of smart radiotherapy biomaterials (SRBs) for combining radiotherapy and immunotherapy. These SRBs include smart fiducial markers and smart nanoparticles made with high atomic number materials that can provide requisite image contrast during radiotherapy, increase tumor immunogenicity, and provide sustained local delivery of immunotherapy. Here, we review the state-of-the-art in this area of research, the challenges and opportunities, with a focus on in situ vaccination to expand the role of radiotherapy in the treatment of both local and metastatic disease. A roadmap for clinical translation is outlined with a focus on specific cancers where such an approach is readily translatable or will have the highest impact. The potential of FLASH radiotherapy to synergize with SRBs is discussed including prospects for using SRBs in place of currently used inert radiotherapy biomaterials such as fiducial markers, or spacers. While the bulk of this review focuses on the last decade, in some cases, relevant foundational work extends as far back as the last two and half decades.

## 1. Introduction

Current cancer treatments include radiotherapy, surgery, chemotherapy, and more recently, immunotherapy. Radiotherapy (RT) is used in the treatment of over 50% of cancer patients. While the options available are expanding and improving, there are still many factors limiting treatment. These include tissue-specific toxicities from radiation and systemic toxicities from chemo- and immune-therapy, which can include long-term side effects such as neuropathy, cognitive problems, kidney damage, hearing damage, and other morbidities [1,2,3].

Increasingly, radiotherapy is being investigated in combination with immunotherapy [4], sometimes called Radio-Immunotherapy. Such combination approaches may increase toxicities that can be severe or even life threatening [5]. Currently, inert radiotherapy biomaterials, such as fiducials and beacons, are used during image-guided radiotherapy to minimize local toxicities, allowing radiation to be more targeted to the tumor while minimizing the irradiation of healthy tissue. The use of smart radiotherapy biomaterials (SRBs) or biomaterial drones provide a major opportunity for achieving the same function as well as minimizing toxicities when combining radiotherapy with immunotherapy. These SRBs can be produced by customizing currently used single-function biomaterials, e.g., fiducials, designed to provide image-guidance during RT. This customization upgrades the technology to be multifunctional, allowing SRBs to provide image-guidance and also serve as radiosensitizers, immunoadjuvants, and/or a vehicle for drug delivery [6]. These upgrades can be facilitated via functionalization by way of Poly(lactic-co-glycolic acid) (PLGA), chitosan, or even carbohydrate polymers. Additional benefits of the latter two include pronounced anti-microbial properties to further enhance treatments [7]. Such technology has the potential for localized delivery of immunotherapy targeted to the tumor microenvironment to minimize systemic toxicities. The use of SRBs is particularly exciting for combining radiotherapy and immunotherapy to generate an in situ vaccination against tumor antigens. An important benefit to creation of the in situ vaccination is lowered barriers to care. While traditional cancer vaccines take weeks to create with a hefty price tag (upwards of USD 36,000 for a single treatment), SRB mediated in situ vaccination would remove the waiting time and could decrease cost due to delivery method optimization [8,9,10].

In situ vaccination with radiotherapy has been described as the abscopal effect, whereby radiotherapy of a tumor at one site may lead to the regression of tumors that are not treated, such as distant metastases [5]. Strategic use of SRBs for in situ delivery of immunotherapies has significant potential to enhance in situ vaccination. Successful clinical translation of SRBs would extend radiotherapy for curative treatment of both local and metastatic diseases that are responsible for over 90% of cancer deaths [11]. Here, we review the development of radiotherapy biomaterials, and the potential merits of SRBs for radio-immunotherapy combining image-guided radiotherapy and immunotherapy. We present a potential roadmap to clinical translation and opportunities for other research and development, including new radiotherapy approaches, such as FLASH.

## 2. Radiotherapy Biomaterials

### 2.1. Currently Used Radiotherapy Biomaterials

Different types of biomaterials have been developed for image-guided radiotherapy or ensuring beam-to-target geometric accuracy and precision targeting during radiotherapy treatment, including motion management. These biomaterials employ various types of fiducial markers, beacons, and spacers. Fiducial markers include metallic seeds, coils or even liquid to provide image contrast across different modalities. In the past, beacons have also been used to provide guidance by transmitting low level RF signals that can be tracked in real time using specialized equipment. Due to the nature of liquid versus solid forms, the liquid fiducial may be easier to administer [12,13]. The additional advantages of liquid fiducials include minimizing migration and image artifacts [14,15,16,17]. Brachytherapy spacers, on the other hand, are primarily used to space out the radioactive seeds to ensure placement accuracy and facilitate dosimetry [18].

### 2.2. Smart Radiotherapy Biomaterials

In a previous review we have described the development of smart radiotherapy biomaterials, which can serve the same functions as their inert counterparts described above but have additional capabilities [6]. Their design allows the SRBs to provide image-guidance as well as other functions, such as radiosensitization, modifying the tumor microenvironment, and sustained delivery of payloads to enhance treatment efficacy. The payloads can include chemotherapy, immunotherapy or nanoparticles that can increase damage to tumor cells during radiotherapy [19]. The specific use of high atomic number (high-Z) nanoparticles provide enhanced image contrast while also enabling photon-induced emission of secondary electrons, to deliver additional damage to the targeted cancer cells within a few micrometers [20]. Examples include gold nanoparticles (GNPs), and gadolinium nanoparticles (GdNPs), which can provide both MRI and CT contrast [21]. In the previous review the different types of SRBs were described [6]. Here, we focus on more recent developments, with specific focus on SRBs for combining radiotherapy and immunotherapy for in situ vaccination.

In the past, cancer vaccination has relied on external processing of tumor cells or dendritic cells harvested either directly from a patient or engineered in a lab to elicit a tailored anti-tumor immune response [22,23]. This anti-tumor immune response is evoked by the presentation of these cells to the immune system, thus training the immune system on what to target [24]. SRBs provide another approach for vaccination, highlighted in Figure 1, where image-guided RT using SRB generates neoantigens unique to each tumor that are taken up by dendritic cells, which migrate to draining lymph nodes and activate antigen specific T cells. The SRBs can also sustainably deliver immunotherapy into the tumor microenvironment to boost the in situ vaccination at different levels, providing a multi-pronged strategy that can maximize both local and metastatic tumor kill. 

### 2.3. Seed Smart Radiotherapy Biomaterials

The first types of SRBs under development are seed-like and are similar to solid fiducials and brachytherapy spacers. Figure 2 illustrates the design of these seeds to provide image contrast and radiation enhancement as well as delivery of immunotherapy. The seed SRB can be created with a mix of PLGA polymer and formed to the size and shape of current fiducials [25]. As can be seen in Figure 2, high-Z nanoparticles can either be loaded together with drugs and incorporated in the hollow core of the SRB (A) or the nanoparticles can be incorporated within the biodegradable polymer matrix in the annulus of the cylinder (B). 

The hollow SRB is advantageous because it allows for direct loading of drugs in the core with sustained release as the polymer degrades. In this way, both NPs and drugs have a constant presence of the drug within the tumor sub volume, which is thought to help overcome immunosuppression, as discussed in recent work [25]. The same investigators highlight the possibility of the biomaterial supporting maturation of dendritic cells and further include animal studies demonstrating that this approach can substantially prime the abscopal effect [25]. While the results support the use of the seed SRB to elicit the immune response for both treated and metastatic tumors, the authors acknowledge the necessity of improved optimization. This is in line with other author’s work citing the necessity for optimization of immunotherapy drug choice and dose as well as concomitant RT dose, fractionation, and selection of relevant high Z NPs [21,25,26,27].

### 2.4. Liquid Smart Radiotherapy Biomaterials

The second type of SRB under development is in the form of a liquid fiducial, called liquid immunogenic fiducial eluter (LIFE) Biomaterial [28]. The LIFE SRB technology is composed of a solution of sodium alginate (ALG) and chitosan that is rapidly transformed into an anchored hydrogel in the presence of calcium ions (Ca^2+^) within the tumor, illustrated in Figure 3 [28,29]. Immunotherapy drug payloads such as anti-CD40 and/or high-Z nanoparticles such as gold (GNP) or gadolinium nanoparticles can be incorporated into this biodegradable polymer to expand it from single function to multi-functional technology [28]. This enables a potent amount of the immunotherapeutic agent to diffuse locally for a sustained period within the tumor with minimal leakage into neighboring organs. The LIFE SRB represents a new generation of multifunctional liquid fiducials, combining image-guidance with in situ drug delivery, to engender effective in situ vaccination. This has the potential for extending the role of RT from local palliation to lasting local and distant disease control of metastatic tumors, such as cervical cancer. Preclinical studies have demonstrated that both MRI and CT contrast gradually decrease over the span of 3 weeks, corresponding to the biodegradation of the LIFE biomaterial [28]. The study also notes significant decrease in tumor progression [28].

Figure 5A highlights results in animals showing visibility of the LIFE SRB in tumor over time. Figure 5B summarizes tumor control and survival of pancreatic cancer when treated with LIFE SRB loaded with titanium oxide and anti-CD40 with and without radiation versus radiation alone [28]. These studies, as reflected in Figure 4 and Figure 5, corroborate the immune-mediated response and show significant regression of both local treated pancreatic tumors and untreated contralateral tumor, representing metastasis, when using RT in combination with either seed-type (Figure 4) or LIFE gel-type (Figure 5) SRB compared to when treating with RT or RT with anti-CD40 alone [30]. Studies also show that the use of SRB is immunogenic (Figure 4C), significantly enhancing infiltration of APCs that are crucial for priming metastatic tumor kill [30,31]. Altogether, the results demonstrate major potential for SRB technology, especially LIFE gel-type, as a multifunctional fiducial that can provide both image-guidance as currently needed clinically (Figure 4) but also boost local and metastatic tumor kill [28].

### 2.5. Nanoparticle Smart Radiotherapy Biomaterial

The third type of smart radiotherapy biomaterials are nanoparticle drones (nanodrones). Nanodrones loaded with immunotherapy drugs and with a high-Z component can be administered to accumulate in the tumor. Examples include nanodrones made of gadolinium or gold or a hybrid, which permit the tracking of this accumulation via CT or MRI. Such platforms can be optimized for image-guided drug delivery, which could allow for quantification of distribution in tumors over time [33,34].

Some nanodrone platforms are being investigated to target lung cancer [35]. One promising approach for administration of these nanodrones is via inhalation. In inhalation delivery, it is important to know the deposition pattern of the nanoparticles in various generations and parts of the lung (airways and alveoli) as well as the size distribution of the nanoparticles therein. Figure 6 shows the foundational steps taken to apply the dedicated aerosol dynamic computer code, SAEROSA, to determine particle deposition in inner-lung volumes [36]. SAEROSA accounts for coagulation and inter-particle dynamics as well as various deposition mechanisms based on boundary layer theory in confined spaces [37]. When compared to bulk deposition fractions from in vivo experiments, SAEROSA simulations demonstrate good agreements [32,38]. These computations provide detailed insight of generation-wise nanoparticle behavior, including the role of coagulation and changing size distribution. At high initial inhaled nanoparticle concentration, which is required for therapeutic effect, interparticle collisions become non-negligible and give rise to altered size distribution and deposition fractions compared to low concentrations (not shown here). Information of particle deposition patterns and size distribution are instrumental in developing an eventual treatment-planning algorithm.

## 3. Roadmap to Clinical Translation

One approach that is being supported by NIH for clinical translation of SRBs is radio-immunotherapy dose-painting. Radio-immunotherapy traditionally refers to the use of an antibody labeled with a radionuclide to deliver cytotoxic radiation to a target cell. However, as of the late 2000’s, this has come to represent a treatment approach combining radiotherapy and immunotherapy [39,40,41]. In radio-immunotherapy dose-painting, only a sub-volume of the tumor, determined via, e.g., imaging, needs to be irradiated and treated with immunotherapy as delivered by SRBs [31]. In studies conducted in prostate and pancreatic cancer, it was shown that radio-immunotherapy dose-painting using SRBs consistently resulted in effective in situ vaccination [31].

The appeal of this dose-painting approach, highlighted in recent work, is that it may substantially reduce toxicities. Targeting a tumor sub-volume with radiotherapy added to continued release of immunoadjuvant from SRBs minimizes the amount of surrounding healthy tissue to be irradiated and there may be reduced or no need for added margins around the tumor, called the clinical target volume in radiation oncology. Targeted release of immunoadjuvants via SRB allows for a controlled and localized release of immune-adjuvant payloads, thus reducing the systemic distribution and minimizing related toxicities [42].

Other advantages of the SRBs include increased immunogenicity and benefits to hypofractionated radiotherapy (HFRT) as well as high dose rate (HDR) brachytherapy. In situ (intra-tumoral) vaccination with SRB technology may help overcome immunosuppression due to the sustained localization of the immunoadjuvant and simultaneous presence of RT-generated neoantigens in the tumor microenvironment. The use of immunoadjuvants, such as anti-CD40, may increase the activation or maturation of antigen presenting cells that pick-up the neoantigens [28,30]. Delivering immunoadjuvants with SRBs may also increase retention within the tumor, substantially minimizing systemic/overlapping toxicities which are currently a limitation with other approaches [30].

In clinical translation studies, the optimal choice of SRBs for in situ vaccination may depend on the type of cancer and where it is located. This is due to difficulties imposed by the location surrounding the cancer, where unique challenges arise relating to dose limitations and related short- and long-term treatment side effects. In pelvic-area cancers, complications can include infertility, incontinence, and radiation-induced disease, while chest-area cancers can have radiation therapy related complications with motion management, skin irritation and radiation pneumonitis [43]. Most, if not all, of these complications stem from too much radiation delivered to healthy organs, whether from the necessity of proper tumor coverage or due to inability to exactly reproduce patient or tumor positioning at the time of treatment. Table 1 shows proposed matching of specific cancers to different forms of SRB. For example, seed SRBs may be readily employed in the treatment of prostate cancer as they can replace currently used seed fiducial markers. On the other hand, LIFE SRBs may be more appropriate for cervical or breast cancers, head and neck cancers or other tumors where liquid fiducials are used. An important benefit of the LIFE biomaterial is the similarities it has with liquid fiducials, which have been shown to have minimal migration and lessened scatter-induced image artefacts [14,15,16,17,44]. Both of these aspects are important to ensure viability as a dependable source for set up with scatter affecting image clarity and migration affecting relative positioning. 

Nanodrone SRBs have been considered for targeting lung tumors, including via inhalation delivery [35]. In highly sensitive areas, such as the brain and other head and neck cancers, seed or LIFE SRBs may not be optimal for payload delivery, given the route of administration. Further, intratumoral administration of single-function nanoparticles can be too invasive for many head and neck area cancers or their metastases—something that current clinical trials of this nature must be aware of [45]. This barrier could be assuaged with incorporation into multifunctional nanodrones. Alternatively, if looking at high-risk cohorts such as head and neck patients with high cervical nodal metastatic spread, clinicians could target clinically accessible nodes with SRBs versus the tumor of origin to elicit the in situ vaccination effect for distant metastasis. A final cancer to highlight in the translation of SRBs to clinical use is skin cancer. With its favorable location, the intra tumoral administration of SRBs is easier than most other cancers. Further, these lesions have a tendency towards immunogenicity due to their UV-induced DNA damage [46]. This is an especially important aspect to consider when dealing with post-transplant immunosuppressed patients who are at risk of developing skin carcinomas where innovative therapeutics are needed given concerns related to transplant rejection with current checkpoint inhibitors [47].

Previous work using other nanoparticle platforms has demonstrated the use of nanoparticles targeting brain tumors [34]. These studies, focused on the gadolinium-based nanoparticles called AGuIX^®^, found that intravenous injection of AGuIX GdNPs lead to accumulation in the tumor in animal models, serving as a basis for protocol of an initial Phase I human clinical trial [48,49]. Further phase I and II clinical trials have relied on the AGuIX GdNPs as well [50,51,52,53,54,55]. Nanodrone SRBs to deliver immunotherapy can build on lessons from such platforms to add the dimension of immunoadjuvant delivery for in situ vaccination. Immunoadjuvants could include agonists, checkpoint inhibitors or monoclonal antibodies [56,57,58]. Intravenously administered neoadjuvant cemiplimab, for instance, is a monoclonal antibody with high efficacy in treating cutaneous malignancies when combined with curative surgery [59]. If incorporated with an SRB delivery system alongside radiosensitizers, pre-surgical RT could be administered as a primer for the immunoadjuvant, potentially allowing to eliminate the need for aggressive or radical surgical resection.

Given this potential of SRBs for combining radiotherapy and immunotherapy, there is rationale for considering combination of SRBs with FLASH Radiotherapy (FLASH RT) technique as an approach for clinical translation. FLASH RT involves treating target volumes at dose rates significantly higher than current standard practice, e.g., >40 Gy/s, which minimizes damage to normal tissue [60]. The potential for FLASH RT to minimize normal tissue toxicity has already been shown [61,62,63]. This approach has garnered significant recent attention due to its normal tissue sparing effects and represents a possible avenue for RT dose-escalation or dose-painting that merits further investigation [64]. Moreover, recent work has suggested that FLASH RT may also have immunologic implications, with two recent reports suggesting that FLASH RT can upregulate T-cell activating and trafficking markers in both glioblastoma and diffuse pontine glioma models [65,66]. 

## 4. Perspective Discussion

To optimize the use of SRBs, further investigations into the optimal material components are needed. The ideal property of such material is stimulus induced dissolvability that would allow for a slow and continuous release of chosen payload appropriate for the treatment schedule [6]. With the high level of localization provided by the SRBs, more concentrated doses of loaded therapy can be delivered directly to the tumor, avoiding the toxicities related to systemic distribution discussed previously. 

In addition to reducing toxicities related to systemic over distribution of immunoadjuvant payloads, incorporating high-Z NPs, e.g., gold, gadolinium or even well-known chemotherapy drugs, such as cisplatin or carboplatin that contain platinum, into SRBs could bolster damage to tumor cells via the photoelectric effect [6]. In high-Z materials, below about 500 keV incident X-ray energies, photoelectric effect is the dominant interaction mechanism. The photon is absorbed and one or more electrons along with low-energy characteristic X-rays are emitted. The benefit of eliciting the photoelectric effect is that sparsely ionizing radiation (X-rays) are converted to densely ionizing electrons that are emitted by high-Z NPs, which results in enhanced energy deposition (dose) in the locality of the NP [67]. The ability to deliver a tightly localized radiation “boost” dose and concomitant damage to the tumor tissue while keeping the overall dose much lower permits normal tissue sparing [20]. This NP-induced boost dose has been shown to result in higher DNA damage than without [68]. 

With documented benefit to both metastasis and local recurrence alone and increased tumor immunogenicity when combined with immunotherapy, high linear energy transfer (LET) carbon ion therapy could also be investigated with the use of SRBs [69]. Carbon ion therapy is a type of heavy ion therapy wherein the use of the spread-out Bragg Peak allows for enhanced dose distribution and increased tissue sparing when compared to conventional RT and has already been suggested as a means to create an in situ vaccine [70]. Proton therapy, another type of heavy ion therapy which operates similarly, has also been shown to synergize with the immune system [70]. SRBs loaded with immunoadjuvant payloads combined with heavy ion therapy may enhance the immune response seen when combining heavy ion therapy with intravenously administered immunotherapy [71].

Use of nanoparticle SRBs may also benefit the development of neutron capture therapy. This could employ nanoparticle SRBs made of gadolinium (Gd) due to the high interaction probability with thermal neutrons [72,73,74,75]. Gd neutron capture therapy (GdNCT) has been discussed in the literature as a means to overcome limitations currently faced by boron-mediated neutron capture, especially with its already established use as MRI contrast with high uptake in tumors [76]. Additional benefits can be seen in the secondary radiations Gd can produce upon neutron interactions. These secondary radiations span photoelectrons, capture gamma rays, X-ray emissions, conversion electrons and Auger electrons which can then be reabsorbed by the Gd to create a cascading effect of further radiations [67,77]. This approach may make the tumor more immunogenic. In preliminary simulations, the dose enhancement factor for GdNCT has been shown to be significant when compared to treatment without nanoparticles (Figure 7) [78]. 

Vaccination approaches are typically dependent on tumor-associated antigens being recognized and picked up by the immune system, and having a sufficiently robust response to be able to overcome the immunosuppression of the tumor microenvironment. Part of the appeal of using the approach of SRBs for in situ vaccination is that the neoantigens generated by RT are done in situ and are specific to the patient’s tumor. This could provide another option to current clinical trials whose focus is on personalized vaccination created ex situ in a lab environment [79,80,81,82,83]. The use of immunogenic SRB components such as polymers can enhance recruitment of dendritic cells to pick-up the antigens. The sustained delivery of immunoadjuvants such as anti-CD40 can then bolster the activation of dendritic cells for more robust in situ vaccination. These approaches, both with standard therapy combinations as well as novel ones, all serve to remove barriers to care for patients, especially from economically disadvantaged areas from rural united states to low and middle income countries [84,85,86].

More work still needs to be done in order to implement and optimize these approaches [25]. Optimization will include the type of radiation and radiotherapy approach, nanoparticle type, size, and concentration, and type and dose of any immunoadjuvant for different cancers as some cancers may be more resistant [48,87]. In addition to therapeutic dose considerations, optimization is needed for the SRBs on programming sustained delivery timing and formulations.

## 5. Conclusions

SRBs technology development represents a major opportunity to provide the next generation of radiotherapy biomaterials or fiducial markers that are multi-functional, enabling image-guidance during radiotherapy, but also boosting in situ vaccination. The potential for using SRBs for image-guided immunotherapy delivery, hence combining radiotherapy and immunotherapy, has shown to be promising in preclinical studies. Successful clinical translation would extend the use of radiotherapy to curative treatment of both local and metastatic disease. The use of innovative approaches such as radio-immunotherapy dose-painting could help minimize systemic and overlapping toxicities hence also enhancing the quality of life of patients. More studies optimizing the use of SRBs for different cancers are an important area of investigation. This is especially true when considering the possible growth into novel combinations such as HFRT, FLASH RT, neutron capture and proton beam therapies.

## Figures and Tables

**Figure 1 nanomaterials-13-01844-f001:**
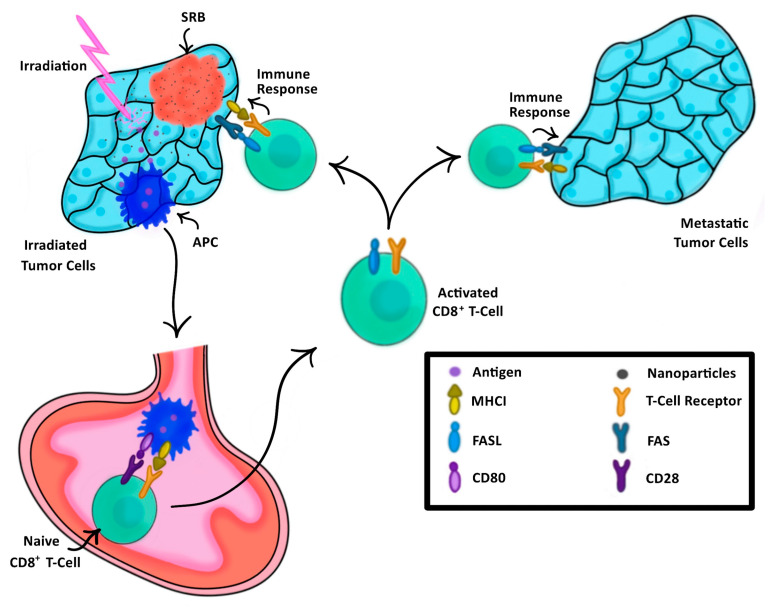
The process of SRB-assisted abscopal effect wherein an initial tumor (**left**) is treated with both radiation enhancing NPs and irradiation, triggering a cascade of antigen-presenting cells (APCs) to the lymph node (**bottom**) to activate naïve CD8+ T-Cells, thus triggering an immune response not only to the initially treated tumor, but also to the metastatic tumor (**right**).

**Figure 2 nanomaterials-13-01844-f002:**
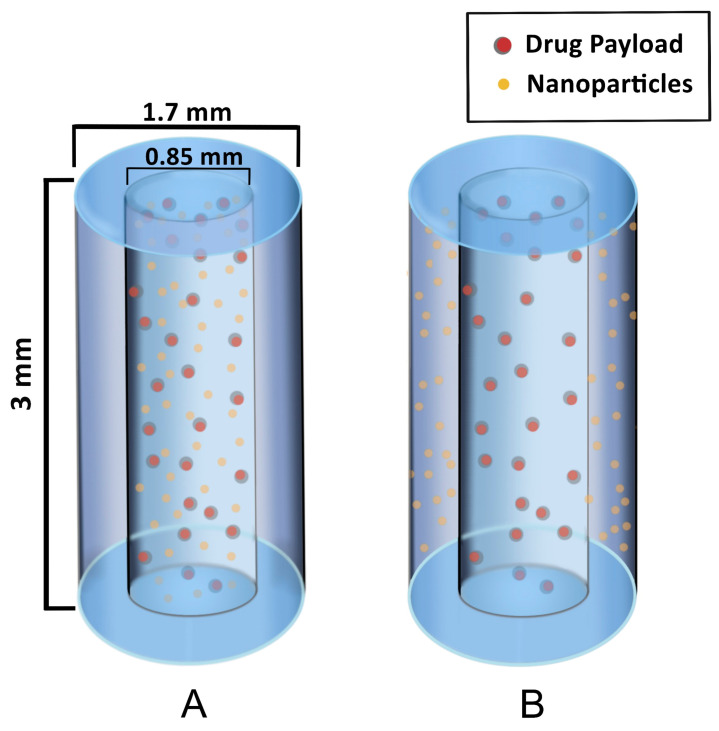
Hollow degradable fiducial SRB loaded with drug payload and nanoparticles. (**A**) One formulation has the NPs within the hollow center, mixed with the drug payload. (**B**) Another formulation has the NPs incorporated within the polymer matrix in the annulus for a quicker release of NP while the drug payload is confined to the core, permitting a similar release rate as in case (**A**).

**Figure 3 nanomaterials-13-01844-f003:**
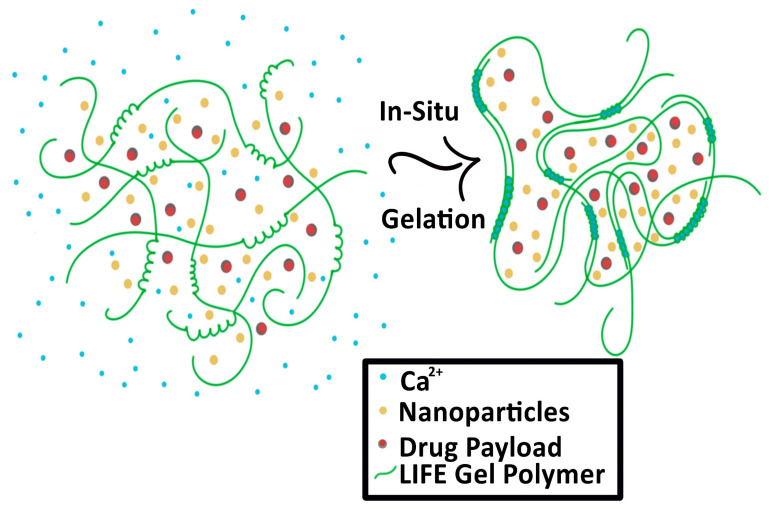
Process of the Liquid Immunogenic Fiducial Eluters (LIFE) Biomaterial loaded with nanoparticles and drug payload as it solidifies in situ. The LIFE biomaterial polymers (green) interact with the free calcium ions (Ca^2+^, blue) and solidify in the tumor.

**Figure 4 nanomaterials-13-01844-f004:**
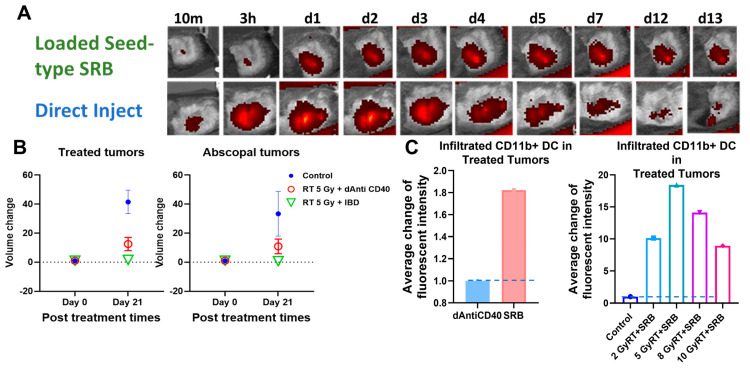
(**A**) in vivo fluorescence imaging (FLI) comparing direct injection of fluorescence-tagged Anti-CD40 with seed-type SRB loaded with Anti-CD40 in mice. The images demonstrate superior sustained presence of antibody when administered with SRB versus direct injection, showing greater presence up to 13 days post-administration. (**B**) Scatter plots of percent volume change of treated and abscopal tumors when treated with IGRT of 5 Gy was given in combination with either direct injection of anti-CD40 or seed-type SRB loaded with the same [30]. (**C**) Bar graph of the average fluorescent intensity of immunofluorescence-stained prostate cancer tissue treated with mouse CD11b+ antibody administered intratumorally vs. via smart radiotherapy biomaterials (SRB) at posttreatment day 7 [31]. Bar graph showing the infiltration of APCs such as dendritic cells (CD11b+) to the treated tumors on day 7 post treatment for varying doses of RT with SRB loaded with mouse antibody versus control [28,32]. Graphs adapted from cited references.

**Figure 5 nanomaterials-13-01844-f005:**
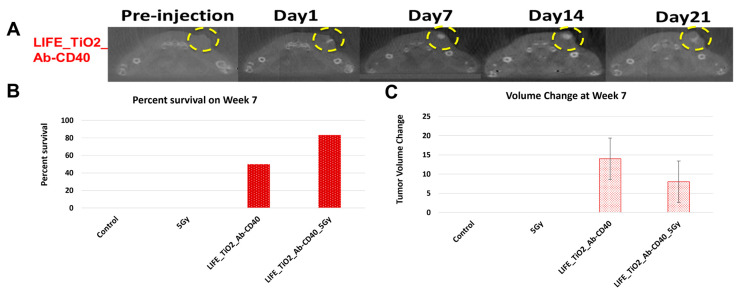
(**A**) CT images showing contrast from LIFE biomaterials injected in mice pancreatic tumors up to 21 days post injection (circled in yellow). Mice were monitored up to 10 weeks after treatment with combinations of RT and LIFE gel, loaded with titanium oxide and anti-CD40. The survival fraction (**B**) and overall change in tumor volume (**C**) for week 7 are shown in snapshot here demonstrating better tumor control when combining SRB LIFE gel with radiotherapy [28]. Graphs adapted from cited references.

**Figure 6 nanomaterials-13-01844-f006:**
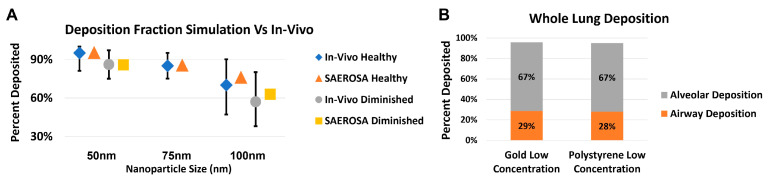
Preliminary results validatinging the application of the aerosol dynamics computer code SAEROSA in a detailed lung model [36]. Simulation results are compared to published in vivo bulk deposition results for polystyrene nanoparticles [38]. Graph (**A**) shows good agreement between simulated versus experimental bulk deposition for polystyrene nanoparticles at 50, 75 and 100 nm. Concentrations, particle sizes and materials used in simulations reflected those used in vivo. Graph (**B**) demonstrates the first step towards therapeutic applications by comparing the deposition within the airway and alveoli separately for both polystyrene and gold nanoparticles, both with 50 nm and experimental concentrations. In-vivo data obtained from cited reference.

**Figure 7 nanomaterials-13-01844-f007:**
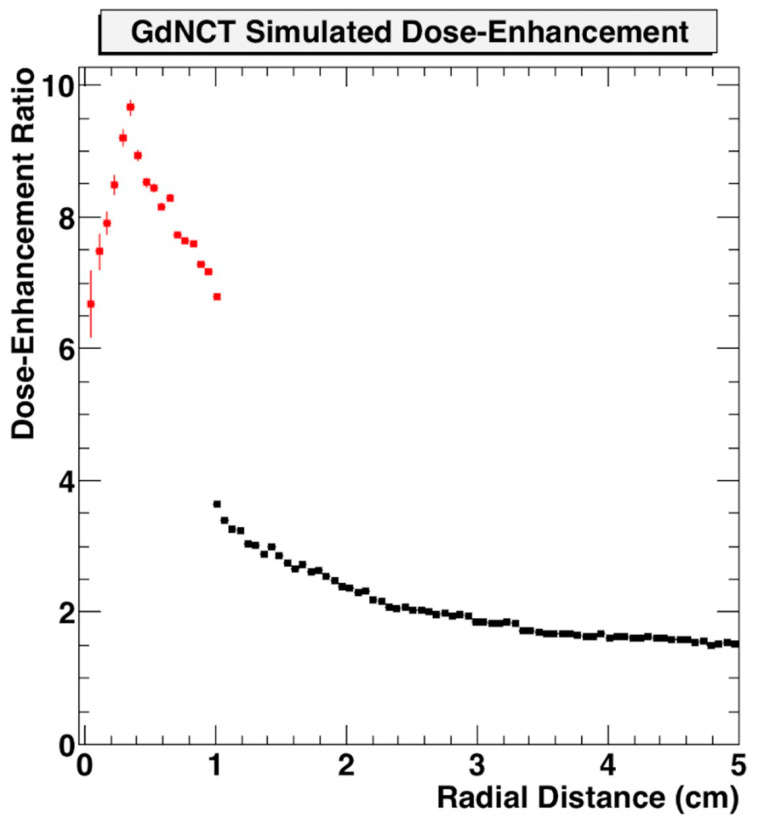
Geant4 Monte Carlo simulation of dose enhancement due to neutron-activated Gd emissions compared to absorbed dose from neutrons alone as a function of distance from the tumor centroid inside the tumor volume (red) and outside the tumor volume (black) [78]. Figure reproduced from conference proceedings with permission from author.

**Table 1 nanomaterials-13-01844-t001:** Comparison of various cancer sites and their most likely form of in situ vaccination. Each site has unique attributes that lend towards one methodology over another.

Type of Cancer	Best In Situ Vaccine Treatment
Brain	Nanoparticle drone SRBs
Breast	LIFE SRBs
Thoracic	Seed SRBs
Pancreatic	Seed or LIFE SRB
Cervical	LIFE SRB
Lung	Seed or Nanoparticle drone SRB
Prostate	Seed or LIFE SRB

## Data Availability

Not applicable.

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
