# Peer review of "Smart Radiotherapy Biomaterials for Image-Guided In Situ Cancer Vaccination"

_nanomaterials, 2023, doi:10.3390/nano13121844_

Round 1

Reviewer 1 Report

This review topic is interesting and the authors have provided a good summary about this topic. Reasonable revisions are needed before acceptance.

1, an illustration figure is suggested

2, more figures/tables are suggested 

3, at the beginning, a few more sentences about biomaterials are suggested, some refs could be read or cited, such as: Chemical reviews,2016, 116 (4), 2602-2663; Carbohydrate Polymers,2021, 266, 118097

4, the significance of this topic need more discussions 

NA

Author Response

Addressing Points 1+2:

  • I split the old Figure 4 into two separate figures to accommodate the addition of a new in-vivo image. The two new images are Figures 3 and 5.  Figure 3(line 143) takes old Figure 4 C-D and combines with the new in-vivo image (3A).  Figure 3 was placed within the “Seed Smart Radiotherapy Biomaterials” section while Figure 5(line 179) is placed within the “Liquid Smart Radiotherapy Biomaterials” section.

Addressing Point 3:

  • Lines 51-54 reflect two new sentences about upgrading biomaterials, including functionalization of the materials via various polymers including the approach in the suggested reference.

Addressing Point 4:

  • Lines 58-62:Added some sentences and references during introduction section to later connect to an added sentence + references in perspectives discussions to highlight the significance and impact on removing barriers of care.

Document was also double-checked for spelling and grammar mistakes as per the request for "minor editing of English language"

Reviewer 2 Report

This manuscript highlights the challenges and opportunities on smart radiotherapy biomaterials for image-guided in-situ cancer vaccination. Authors have shown a particular focus on in-situ vaccination to expand the role of radiotherapy in the treatment of both local and metastatic disease. They have also presented a roadmap for clinical translation. The manuscript could be accepted after minor revisions.

1. Mention time duration of literature covered in the abstract.

2. Figures do not show any copyright information if they are adapted.

3. Future perspective should be added describing how the author's see this field growing?

4. Some inconsistencies in the font, presentation style are noted. Adhere to the journal's instructions for figures and tables.

5. In the reference section, some journal names are abbreviated while others are not. They should be consistent and follow journal's format.

Author Response

Addressing Point 1:

  • Lines 26-27: Added a sentence in the abstract with statement of what time duration the bulk of work is focused on as well as mentioning the farther reaching timeline of some foundational background information

Addressing Point 2:

  • Added a line in figure caption for Figs 3&4 that graphs were adapted from cited references.(lines143;179 respectively)
  • Added a line in figure caption for Fig 6 to reflect that graph was reproduced from conference proceedings with permission from author.(Line 364)

Addressing Point 3:

  • Added a sentence in the conclusions to highlight the possible growth into areas discussed in “perspective discussion”. I am somewhat unsure how to further expand what we already have in “perspective discussion” to more explicitly state the growth of the field? (Lines 395-398)

Addressing Point 4:

  • I believe I have fixed the font issues within the text. Using the Nanomaterials Word Template, I have used the pre-loaded styles for the corresponding sections including figure captions.

Addressing Point 5:

  • Addressed and fixed in all references.